# And the credit goes to . . . - Ghost and honorary authorship among social scientists

**Gernot Pruschak** [1,2,3]*, **Christian Hopp** [1,3]

1 TIME Research Area, RWTH Aachen University, Aachen, Germany, 2 Department of Business Decisions and Analytics, University of Vienna, Vienna, Austria, 3 Institute for Applied Data Science & Finance, Bern University of Applied Sciences, Bern, Switzerland

* gernot.pruschak@univie.ac.at

**Data Availability Statement:** The data cannot be made publicly available due to them containing sensitive information on respondents' authorship malpractices in previous publications. While the data is anonymized, certain combinations of

## Abstract

The proliferation of team-authored academic work has led to the proliferation of two kinds of authorship misconduct: ghost authorship, in which contributors are not listed as authors and honorary authorship, in which non-contributors are listed as authors. Drawing on data from a survey of 2,222 social scientists from around the globe, we study the prevalence of authorship misconduct in the social sciences. Our results show that ghost and honorary authorship occur frequently here and may be driven by social scientists' misconceptions about authorship criteria. Our results show that they frequently deviate from a common point of authorship reference (the ICMJE authorship criteria). On the one hand, they tend to award authorship more broadly to more junior scholars, while on the other hand, they may withhold authorship from senior scholars if those are engaged in collaborations with junior scholars. Authorship misattribution, even if it is based on a misunderstanding of authorship criteria rather than egregious misconduct, alters academic rankings and may constitute a threat to the integrity of science. Based on our findings, we call for journals to implement contribution disclosures and to define authorship criteria more explicitly to guide and inform researchers as to what constitutes authorship in the social sciences. Our results also hold implications for research institutions, universities, and publishers to move beyond authorship-based citation and publication rankings in hiring and tenure processes and instead to focus explicitly on contributions in team-authored publications.

## Introduction

"Publish or Perish" characterizes the academic reward system across scientific disciplines. Hiring and tenure decisions depend upon publication metrics, and funding agencies are increasingly employing publications as their main criteria [1–5]. For academics, competition for publication spots in top-tier journals is fierce [6–8]. Responding to the increasingly complex nature of academic inquiries [9], and the competition in academia, researchers are working more frequently in teams [10,11].

Team-authored academic work allows researchers to increase their productivity by profiting from specialization and the division of labor [12,13]. Bringing together scientists from

demographics might still reveal survey participants' identities. This could compromise survey participants and violate respondents' privacy. Moreover, the GDPR prohibits the public sharing of any data that might lead to the identification of individuals. When providing their informed consent to participate in the study, participants were ensured their privacy would be protected. They did not provide consent for their data to be shared in a repository. Data requests must be addressed to the Data Protection Officer at Bern University of Applied Sciences, who will provide access after the signing of a non-disclosure agreement: datenschutz@bfh.ch.

**Funding:** The authors received no specific funding for this work.

**Competing interests:** The authors have declared that no competing interests exist.

different backgrounds enhances creativity and the depth of the work [14,15], as well as–through mutual checks for potential errors–reproducibility [16]. Nonetheless, the rise of co-authorship also opened the door for new areas of academic misconduct. In team-authored work, it has become possible to withhold authorship from heavily engaged contributors (ghost authorship) and/or to award authorship to someone who didn't participate in a research project (honorary authorship) [17]. Given that authorship misconduct may obfuscate from inferring individual contributions, wrongly assigned co-authorship can distort individual credit.

In a 2005 study, Martinson, Anderson and de Vries [18] found that 10% of the researchers they surveyed had assigned authorship inadequately at least once in their career. Across disciplines, scientists perceive authorship misconduct to be ten times more likely to happen than data fabrication or falsification [19]. While individual instances of authorship misconduct are less damaging to science than transgressions such as data fabrication or falsification, this type of misconduct seems to be much more widespread, and it does have consequences: ghost authorship and honorary authorship distort citation and publication counts, which, as noted above, are among the primary metrics of academic productivity [20].

Therefore, it is not surprising that to prevent adulterations of academic rankings, many highly ranked journals, especially in the natural and life sciences, now require so-called contribution disclosures in which each contributor discloses his or her specific contribution to the paper [21,22]. In 2018, Elsevier, a top publisher of social scientific research [23], adopted guidelines to encourage [24] authors to employ the "CRediT" system [25] for contribution disclosures. CRediT, which stands for Contributor Roles Taxonomy, is a system that breaks down the various roles a contributor can play in writing a paper; contributors can be listed, then, according to the precise roles they played, including conceptualization, methodology, software, writing, reviewing and editing. In turn, some social scientific journals have, begun to encourage authors to submit statements on what role each author played in the composition of a submitted article, but, unlike the journals in the natural and life sciences, they do not require these statements [26–28]. In addition, the ethics guidelines of the largest social science research societies do not even mention contribution disclosures [29–32].

The lack of mandatory authorship contribution statements in the social sciences leads to our research question: How prevalent are ghost authorship and honorary authorship in the social sciences? To address this question, we provide clear and comprehensive facts on the prevalence, distribution, and motivational factors of ghost and honorary authorship in the social sciences.

## Authorship

To adequately assess publication counts and to attribute citation counts to individual authors, it is important to recognize contributions made in team-authored publications. Yet the prevailing definitions of authorship are broad and imprecise because of the wide variety of academic fields they need to apply to [33]. As a case in point, the prevailing modus operandi would identify scholars as authors if they made "substantial contributions" [34] to the publication. Admittedly, this definition of authorship is vague and leaves many degrees of freedom for the scholars involved. Consequently, there is a wide discrepancy between what authors should ideally attribute and what authors do attribute.

In the social sciences, this discrepancy is further exemplified by the authorship definitions of large societies whose codes of ethics or conduct state that individuals who contributed substantially to a publication should receive authorship [29–31]. Regrettably, the definition of a substantial contribution remains unclear in those guidelines. To solve this issue, editorial policies at an increasing number of prominent interdisciplinary journals that also publish social

science articles have begun to implement more specific guidance on who should receive authorship and who should not. These guidelines are either based on [35,36], or very similar to [37,38], the authorship criteria defined by the International Committee of Medical Journal Editors (ICMJE) [39] (the authorship guidelines of *Nature* [37] and *PNAS* [38] are more relaxed as they only require responsibility for submission and one other task). According to the ICMJE, "Authorship credit should be based only on 1) substantial contributions to conception and design, or acquisition of data, or analysis and interpretation of data; [AND] 2) drafting the article or revising it critically for important intellectual content; AND 3) final approval of the version to be published" [39]. The ICMJE revised its criteria in 2013, adding the "agreement to be accountable for all aspects of the work in ensuring that questions related to the accuracy or integrity of any part of the work are appropriately investigated and resolved" [40] as a fourth criterion. Every researcher fulfilling all four requirements is not only eligible for but must receive authorship [40]. It is not only interdisciplinary journals that refer to the ICMJE for details on authorship: the Committee on Publication Ethics (COPE), a multidisciplinary advisory body on ethical issues to which several large academic institutions, journals, and societies subscribe, advises its members to employ the ICMJE authorship standards as well [41]. This highlights the relevance of the ICMJE authorship criteria in the social sciences. Consequently, we employ these criteria as the benchmark definition in our empirical analysis.

## Ghost authorship

Having established the definition of authorship based on the ICMJE criteria, we can elaborate on what constitutes authorship misconduct. The first type we will consider is ghost authorship. A ghost author contributes to a research paper, but, willingly or unwillingly, is not named as an author [42]. Ghost authorship shares with plagiarism the misattribution of intellectual work [43]. Yet, in plagiarism, the originators are usually not aware of someone else stealing their ideas.

Existing research provides three general explanations for the presence and prevalence of ghost authorship. First, pressure from co-authors, can lead to researchers declining or being declined authorship. The rising usage of fractional counts in assessing scholars' productivity rs constitutes a potential reason for this because this motivates researchers to include as few authors as possible to maximize their own publication counts [44]. In such a case, perhaps the ghostwriter is a subordinate of one of the authors, with little power in the relationship [45]. Second, scientists may voluntarily decline author credits because they perceive the findings in the research as controversial, dubious, or weak; perhaps they fear that the publication of the paper may have negative consequences on their future career [46,47]. Third, researchers may evade authorship to disguise potential conflicts of interest. For example, a freelancer sponsored by a pharmaceutical company may approach life scientists with a biased bundle of articles and ask them to write a research paper based on it [48], hoping the paper will promote the official approval, and/or boost the sales, of a drug. As the freelancers do not receive authorship (and thus are ghost authors), the monetary commitment of the firm stays secret [49].

## Honorary authorship

In contrast to leaving out contributors in published work, authorship misconduct may also result from adding individuals to publications that they have contributed to either insubstantially or not at all. This behavior is called honorary authorship [50], gift authorship, or guest authorship [51]. It is beneficial to those receiving it because they can get the credit for the publications without exerting the effort and time needed to conduct research and write articles. As with ghost authorship, pressure and power dynamics explain some of the occurrences of

honorary authorship [52]: for example, senior scientists may demand authorship from their subordinates in return for employing them, from their Ph.D. students in return for supervising them, or to either in return for providing funding [53,54]. In other cases, the original authors voluntarily include honorary authors. The inclusion of a well-known researcher into the author list increases the publication chances, especially with single-blind review processes and for high-impact journals [55]. A famous co-author often boosts citations, too, which is beneficial to all authors [56]. Toward these ends, authors might ask well-known experts to co-author their papers without having participated in the research or writing processes [51]. A third type of honorary authorship happens when researchers engage in reciprocal relationships. The general increase in co-authored papers enables researchers to trade co-authorship: a scientist adds only a small, non-authorship-worthy contribution to a research project but still receives author credit; the original author also receives reciprocal co-authorship when the honorary author publishes his or her next paper. This behavior may be especially common within chairs or research groups, where reciprocal proofreading might lead to honorary authorship [57].

## Research questions

In the following, we aim to investigate to what extent these findings may generalize across disciplinary boundaries. We, therefore, extend prior work by examining authorship assignments in a field where journals only recently started encouraging scholars to disclose their contributions, where the number of authors on average is manageable and where there is (to the best of our knowledge) limited prior work on honorary and ghost authorship: the social sciences.

We believe this examination to be valuable as the results from the life and natural sciences may provide little guidance for the social sciences due to the many contextual differences in which publishing takes place. For example, the average acceptance rates of papers submitted to the respective field journals are quite different. In the social sciences, journals publish only around 25% of all submissions [58]. In nature and life sciences, journals publish approximately 40 to 50% of all submissions [59]. In the following, we study the prevalence of authorship misconduct using a large-scale survey of social science researchers.

Subsequently, we conduct exploratory research on ghost and honorary authorship using various possible determinants of authorship misassignment. We look at job positions because supervisor-subordinate collaborations seem susceptible to authorship misconduct [60–63]. We also look at time spent in academia, as it relates to the specific tasks that researchers take on [64] and thus might relate to authors' eligibility according to the ICMJE criteria. (On the subject of tasks, it is worth noting that a recent study surveying "scientists who publish a paper every five days" [65]–which equals 72 or more published papers per year–showed that more than 70% of the respondents did not conduct at least one of the three required tasks of the ICMJE authorship criteria–excluding accountability–in at least every fourth paper.) In addition, we take into account researchers' gender. Female researchers are arguably among the most vulnerable, as credit often tends to go primarily to their male colleagues. This tendency is known as the Matilda effect, named for the suffragist and abolitionist Matilda Joslyn Gage, who wrote about how women have been denied credit for their achievements throughout history [66]. Finally, we look at culture, as it, too, may influence authorship assignments. Salita [67] discussed the possibility that cultural differences might induce perceptions of authorship that could deviate from the ICMJE criteria.

We thus ask the question of what authors are more likely to confirm ghost or honorary authorship in their published team-authored work. In studying the determinants of these types of misconduct, we provide crucial information that can be used by university officials, journal editors, publishers, et cetera, to implement effective countermeasures.

## Materials and methods

### Distribution

To address our research question, we draw on a large-scale survey of social scientists. We designed the survey in spring 2018 using the online survey tool Qualtrics. In May 2018, we asked colleagues for feedback. After incorporating their feedback, we conducted a test run by sending the survey link to 275 scholars who had presented at least one paper at the 2018 European Accounting Association Annual Congress. Data and feedback from the test run showed no need to adapt the questionnaire further. Therefore, the data gathered in the pilot phase is included in the analysis. We did not obtain approval for the survey from an institutional review board (IRB) as the survey was conducted at a point in time where both authors were employed at the RWTH Aachen University. At the RWTH Aachen University, only life science and psychological experimental studies go to the independent ethics committee. The authors contacted the independent ethics committee of the RWTH Aachen University with a request to review the study. Yet the ethics committee replied that they do not provide ethics oversight for empirical social scientific studies. In fact, neither the RWTH Aachen University, nor German state or federal agencies require or offer ethics oversight for empirical research in the social sciences.

To ensure broad dissemination of the questionnaire among researchers from various fields of research in the social sciences, we selected corresponding authors of published articles in well-known journals as well as of papers presented at conferences organized by large field-specific research societies between January 2007 and June 2018. S8 Table includes a list of the societies and journals. After deleting duplicates, we had a total of 126,480 unique email addresses. A random selection of half of these addresses led to an initial sample of 63,240. These scholars received an email containing a brief explanation of the purpose of the study and a Qualtrics URL link to the questionnaire in late August and early September 2018. The link to the survey was the same for everyone to ensure the anonymity of respondents. We ruled out multiple responses by the same individual through IP address restrictions. After sending out the survey, 15,573 emails automatically bounced back due to the email addresses being no longer in use. The contacted sample, therefore, contains 47,697 valid recipients.

### Sample

The dissemination of the survey resulted in 2,817 responses. This constitutes 4.45% of all contacted email addresses and 5.91% of all valid e-mail address. These rates are comparable to other recently conducted non-incentivized online surveys among scientists investigating academic misconduct [68,69]. Moreover, the demographic characteristics of our sample (available in S1 and S2 Tables) are comparable to the summarized database of more than 30 million Zippia.com profiles affiliated as social scientists [70]. For example, the share of women is 33.2% in our study and 37.2% in Zippia.com [70]. Of those who completed the survey, 2,223 were social scientists. We then deleted one respondent with nonsensical answers who claimed to have been working in academia for 100 years despite being only 67 years old. Therefore, our original sample consists of 2,222 respondents. This corresponds to a response rate of 3.51% among all e-mail addresses and 4.66% among valid email addresses. Our drop out rates are comparable to existing research on academic misconduct [71]. Nevertheless, we investigate whether these dropouts induced potential sample selection bias in the robustness checks. However, the samples used for the following analyses differ because some conference attendees had not yet published in journals and some respondents selected "N/A" for one or more responses. Respondents who had not published a journal paper did not receive the questions

on their last published paper but were still able to respond to two vignettes in the last section that described projects and asked respondents to assign authorship to the researchers involved. Respondents stating that their last published paper did not include any non-author contributors were excluded when comparing the actual and perceived rate of ghost authors in model 5 of Table 1 because of divide by zero errors.

### Survey design

The survey consists of three parts. In the first section, respondents answered questions about their demographic and job characteristics. The second part of the survey asked about the distribution of authorship and contributor acknowledgments in the latest published paper that names the respondents as authors. The third section covered two vignettes, as noted above, which described research projects and asked respondents to assign authorship to the researchers involved.

**Prevalence of authorship misconduct.** To elicit the extent of ghost and honorary authorship, we asked respondents regarding the number of authors and the number of other people, excluding peer reviewers, who had contributed to their last published paper. Following Mowatt et al. [72], we asked respondents to specify for each of the authors and contributors (or for the top five, where there were more than five authors or contributors) whether that person participated in *creating the research design*, *searching for literature*, *analyzing the literature*, *collecting and/or preparing data*, *describing the results*, *writing up the paper*, *reviewing and commenting on the written paper* and *approving the final version of the paper*. According to the ICMJE authorship criteria, only a person *approving the final version of the paper*, *writing up the paper*, and/or *reviewing and commenting on the written paper* and engaged in at least one of the other tasks mentioned should receive authorship (for the sake of simplicity and comparability with existing research, we refer to the original three ICMJE authorship criteria without the recently added criterion of accountability throughout the article) [39]. Based on these criteria, we identified as ghost authors those scholars who fulfill the ICMJE requirements but were not listed as authors, and as honorary authors those who do not fulfill the ICMJE requirements but were listed as authors. Furthermore, respondents were asked to indicate on a scale from 0 (disagree) to 100 (agree) whether they agree that for their last published paper, all researchers who made significant contributions were named as authors. We employed this question as an indicator of respondents' perceptions of the possible presence of ghost authors. The last question in the second part of the survey asked respondents to indicate on a scale from 0 (disagree) to 100 (agree) that for their last published paper, researchers received authorship only if they participated actively in the creation process. In this way, survey recipients stated their perceptions of the possible presence of honorary authors.

**Determinants of authorship misconduct.** In addition to determining the prevalence of authorship misconduct, we also aim to provide exploratory evidence regarding potential determinants of authorship misconduct. As discussed in the introduction, there are a wide variety of potential determinants of authorship misconduct. The questionnaire, therefore, included demographic questions on gender, age, and geographical region. Moreover, we asked respondents to indicate their primary research field, their job position, and how long they had worked in academia. In addition, we asked about respondents' productivity, measured by the number of articles they had published in the three years leading up to the date of the survey, and their level of academic engagement, measured by whether they held a position in an editorial board as well as the number of reviews they had written in the year leading up to the survey date.

**Assignment of ghost and honorary authorship.** The assignment of ghost and honorary authorship in two vignettes was intended to help us further investigate the social scientists' conceptions and misconceptions about authorship. We developed the following two hypothetical

**Table 1. Regression results of actual and perceived prevalence of ghost and honorary authorship.**

| | 1 Ghost Authorship | 2 Honorary Authorship | 3 # of Ghost Authors | 4 # of Honorary Authors | 5 Perceived Ghost Authors | 6 Perceived Honorary Authors |
|---|---|---|---|---|---|---|
| Rate of Ghost Authors | | | | | 0.714 | |
| | | | | | (0.392) | |
| Rate of Honorary Authors | | | | | | 0.935 |
| | | | | | | (0.176) |
| Female | -0.324 | 0.175 | -0.345 | 0.109 | 0.151 | 0.054 |
| | (0.302) | (0.106) | (0.331) | (0.070) | (0.209) | (0.121) |
| Anglophone | -1.619 | -0.964 | -1.416 | -0.612 | -0.663 | -0.237 |
| | (0.456) | (0.206) | (0.509) | (0.123) | (0.352) | (0.206) |
| Continental Europe | -1.227 | -0.786 | -1.228 | -0.522 | -0.577 | -0.373 |
| | (0.419) | (0.203) | (0.493) | (0.121) | (0.337) | (0.203) |
| Developing Countries | 0.259 | 0.131 | 0.265 | -0.118 | -0.085 | -0.267 |
| | (0.405) | (0.231) | (0.516) | (0.134) | (0.340) | (0.227) |
| Age | 0.031 | 0.009 | 0.015 | 0.006 | -0.006 | -0.005 |
| | (0.020) | (0.008) | (0.024) | (0.005) | (0.017) | (0.009) |
| Ph.D. Student | 1.268 | 0.513 | 1.520 | 0.290 | 0.012 | 0.204 |
| | (0.440) | (0.204) | (0.526) | (0.122) | (0.340) | (0.194) |
| Professor | -0.201 | -0.287 | 0.017 | -0.221 | -0.087 | -0.171 |
| | (0.343) | (0.128) | (0.386) | (0.085) | (0.256) | (0.150) |
| Editor | 0.245 | 0.067 | 0.084 | 0.113 | -0.001 | 0.058 |
| | (0.307) | (0.121) | (0.344) | (0.080) | (0.243) | (0.143) |
| Years in Academia | -0.009 | -0.011 | 0.007 | -0.010 | 0.007 | -0.008 |
| | (0.021) | (0.008) | (0.024) | (0.006) | (0.017) | (0.010) |
| Published Papers | -0.023 | 0.074 | -0.105 | 0.062 | 0.075 | 0.024 |
| | (0.108) | (0.042) | (0.125) | (0.027) | (0.082) | (0.048) |
| Written Reviews | 0.063 | -0.010 | 0.039 | -0.020 | -0.164 | -0.023 |
| | (0.096) | (0.036) | (0.110) | (0.024) | (0.084) | (0.042) |
| Business | -0.397 | 0.216 | -0.363 | 0.074 | 0.181 | -0.050 |
| | (0.399) | (0.166) | (0.466) | (0.109) | (0.311) | (0.182) |
| Economics and Finance | -0.515 | -0.430 | -0.560 | -0.395 | -0.110 | -0.278 |
| | (0.500) | (0.199) | (0.597) | (0.139) | (0.399) | (0.235) |
| Computer and Statistics | 0.056 | 0.319 | 0.328 | 0.150 | -0.267 | -0.183 |
| | (0.407) | (0.188) | (0.485) | (0.120) | (0.400) | (0.213) |
| Political Sciences | -0.967 | -0.625 | -1.294 | -0.523 | 0.129 | -0.584 |
| | (0.670) | (0.222) | (0.792) | (0.157) | (0.427) | (0.282) |
| Psychology | -0.410 | 0.321 | -0.060 | 0.338 | -0.311 | 0.056 |
| | (0.676) | (0.240) | (0.693) | (0.150) | (0.548) | (0.256) |
| Sociology | -0.334 | -0.241 | 0.247 | -0.150 | 0.280 | -0.132 |
| | (0.605) | (0.226) | (0.640) | (0.152) | (0.427) | (0.256) |
| Chi-Square | 55.30 | 145.69 | 44.53 | 140.49 | 23.68 | 70.45 |
| P > Chi-Square | 0.00 | 0.00 | 0.00 | 0.00 | 0.1659 | 0.00 |
| Pseudo R-squared | 0.01 | 0.06 | 0.06 | 0.03 | 0.04 | 0.04 |
| Observations | 1854 | 1857 | 1854 | 1857 | 804 | 1818 |

1 and 2 present marginal effects derived from logistic regressions with standard errors in parentheses. 3 and 4 present coefficients derived from negative binomial regressions with standard errors in parentheses. 5 and 6 present coefficients derived from Poisson regressions with standard errors in parentheses. The (omitted) reference categories for the categorical variables are: *Asia* for geographical region, *Postdoc* for job position and *General Social Sciences* for research field. S5 Table includes confidence intervals, p-values and more information on the varying number of observations.

scenarios capturing different authorship team compositions that were involved in the development of research papers. Ideas for these vignettes were derived from [61–63], though our research focus and context differs from these studies. Respondents read each vignette and afterward had to decide whether to award authorship to each individual mentioned in the vignettes.

The first vignette described a collaboration between a postdoc and a professor where both contributed to a similar extent towards the publication, while a student assistant helped in the data collection process. According to the ICMJE authorship criteria, the professor and the postdoc should receive authorship, while the student assistant should not.

For the second vignette, Qualtrics randomly split the respondents into two groups. Respondents in Group A assessed a postdoc/professor collaboration while respondents in Group B assessed a professor/professor collaboration, again with a student assistant helping in the data collection. For both groups, the researcher mentioned first in the second vignette exerted substantially higher efforts than the researcher mentioned second, who reached the minimum threshold of the ICMJE authorship criteria by participating in the conception, reviewing, and final approval. The vignettes differ only in the type of collaboration. In neither case should an author credit go to the student assistant.

As researchers' workloads are the same in the first vignette, the comparison between the randomly assigned groups should not result in differences. Yet the workloads differ between the first and second vignette, resulting in possible variations in authorship assignments between the vignettes.

If, however, we additionally observe differences between the groups in the second vignette, these would be attributable to the perception of rank differences between postdoc/professor and professor/professor collaborations. Thus, we can elicit authorship judgments by the respondents as well as whether these judgments are affected by subjective perceptions about the roles and academic positions of the involved researchers.

## Statistical analysis

We conducted the entire data analysis using Stata 16. For the analyses of authorship assignments in respondents' last published papers, our first set of dependent variables indicates whether the author list of a paper suffers from *Ghost Authorship* and/or *Honorary Authorship*. Our second set of dependent variables employs the exact number of *Ghost Authors* and *Honorary Authors* per paper. Our third set includes the indications of the extent of *Perceived Ghost Authors* and *Perceived Honorary Authors* in respondents' last published papers. For the analyses of the vignettes, our binary dependent variables employ respondents' authorship assignments by each individual respondent.

We employ different empirical estimation strategies for each set of dependent variables. We apply logistic regressions for the dichotomous variables of *Ghost Authorship* and *Honorary Authorship*. For *Ghost Authors* and *Honorary Authors*, we employ negative binomial regressions due to the overdispersion of the count variables. We utilize Poisson regressions for the equidispersed perception-based dependent variables of *Perceived Ghost Authorship* and *Perceived Honorary Authorship*. Last, we employ logistic regression again for the authorship assignments associated with the vignettes.

## Results

### Actual prevalence of authorship misconduct

Based on the ICMJE authorship criteria, out of 1,878 papers with full data on the authorship tasks, one ghost author participated in the creation of 43 (2.29%) papers and two or more ghost authors participated in the creation of 21 (1.12%) papers (Fig 1A). Moreover, regardless

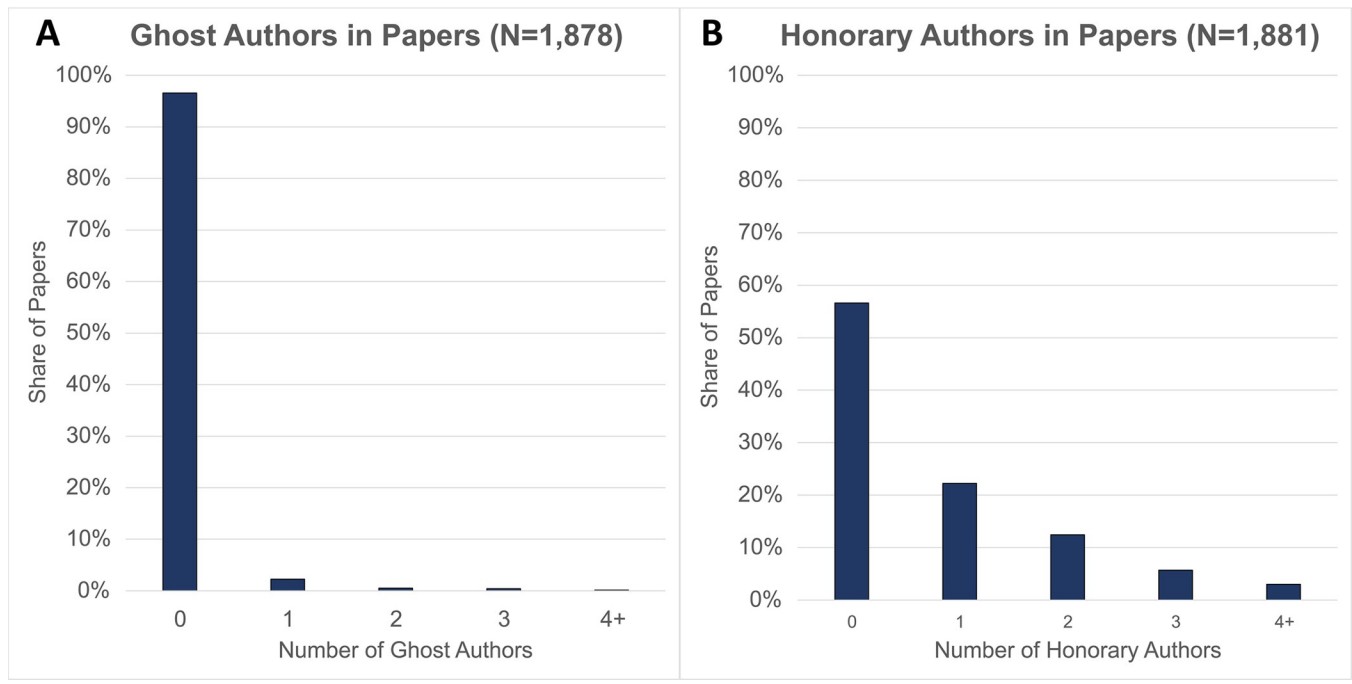

**Fig 1.** Rates of ghost (A) and honorary (B) authorship. N. of obs. are 1,878 for (A) and 1,881 for (B).

of the definition of authorship, we can clearly identify 34 ghost authors who have contributed to all tasks of a research project but did not receive authorship. Honorary authorship, as defined by the ICMJE requirements, occurs much more frequently, with 418 papers (22.22%) containing one honorary author, 234 papers (12.44%) containing two honorary authors, 107 papers (5.69%) containing three honorary authors, and 57 (3.03%) containing four or more honorary authors (Fig 1B). In addition, regardless of the definition of authorship, we can clearly identify 134 honorary authors who did not contribute at all to a research project on which they were named as authors.

We compare the rate of ghost authors that we identified based on the ICMJE guidelines to the degree that respondents believed that authorship was inappropriately withheld from one or more contributors. Fig 2A show that social scientists perceive ghost authorship to be more prevalent than it is: the perceived rate of ghost authorship (light blue) exceeds the identified rate of ghost authorship (dark blue). The difference in the means (4.65% for actual and 13.09% for perceived) is significant according to a two-sided t-test (t = -8.0874; df = 812; p<0.00).

Fig 2B compares the rate of identified honorary authors in each paper to respondents' assessments of the rate of honorary authors in their last published paper. The difference between the identified (dark blue) and perceived (light blue) occurrences is not as stark as for the perception of ghost authorship (the mean of actual is 23.86% and the mean of perceived is 17.26%). Nevertheless, using a two-sided t-test, we find that the perceived rate of honorary authors is, on average, significantly lower than the identified rate of honorary authors (t = 7.5946; df = 1841; p<0.00).

## Potential antecedents to authorship misassignment

To identify the reasons for the mismatches between scholars' perceptions of the prevalence of authorship issues and the actual occurrence of these issues, we investigate their antecedents

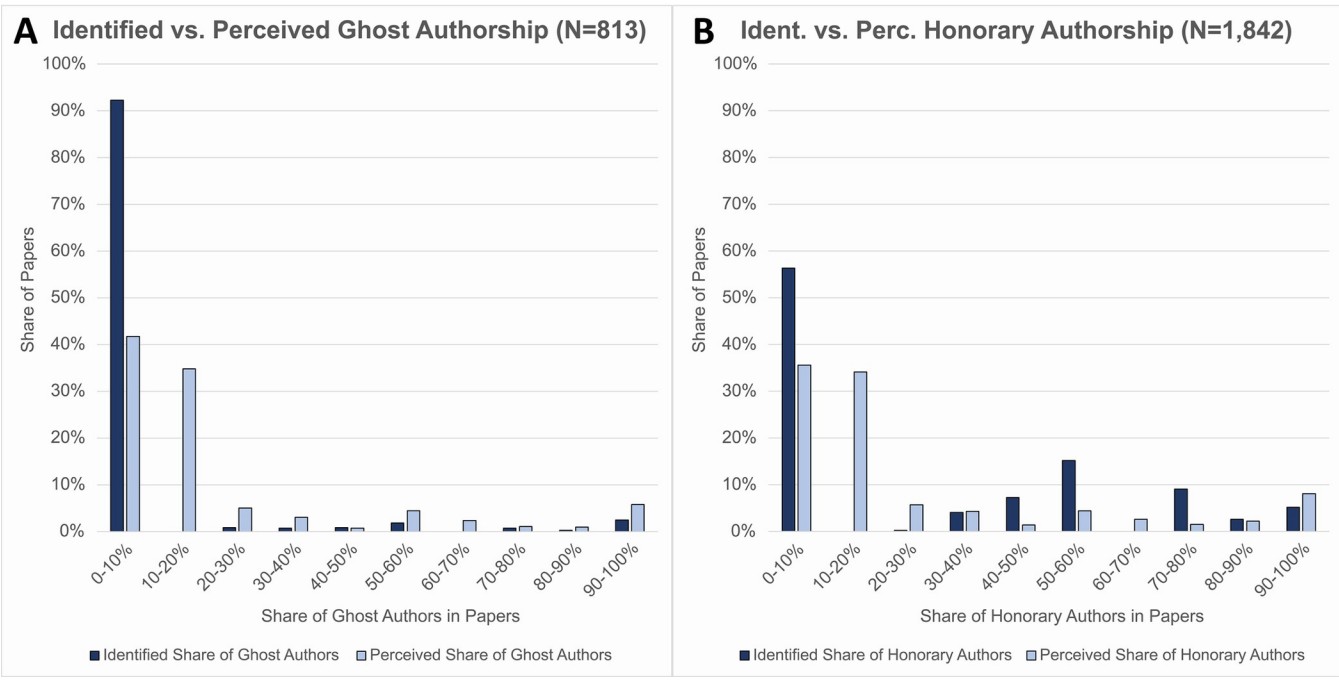

**Fig 2.** Shares of the identified rate of ghost authors and perceived rate of ghost authors (A) as well as the shares identified rate of honorary authors and perceived rate of honorary authors (B). N. of obs. are 813 for (A) (because we only can assess the rate of ghost authors among the papers including at least one non-author contributor) and 1,842 for (B).

and correlates. Table 1 depicts the regression results using a host of explanatory variables. Models 1 and 2 report the results from a logistic regression using a binary dependent variable that takes on the value of one if the researchers' last papers include at least one ghost author (1) or honorary author (2) based on the ICMJE's definition of authorship. Models 3 and 4 report the results from a negative binomial regression employing the number of ghost and honorary authors in each paper as dependent variables.

The results show that women do not exhibit more ghost authorship in their papers than men. Yet while the positive coefficient pointing towards women facing honorary authorship more often in their papers is not significant at the conventional $p < 0.05$ level, this effect is marginally significant at $p < 0.1$ (S5 Table). Furthermore, we find strong effects that scholars in Anglophone and Continental European countries report fewer ghost and honorary authors than scholars from other world regions. Respondents who are Ph.D. students indicate 51.3% more occurrences of honorary authorship and 127% more occurrences of ghost authorship in their last published paper. In contrast, respondents who are professors indicate fewer occurrences of honorary authors but do not differ significantly from the baseline category of junior faculty members regarding ghost authorship. The more papers a scholar has published in the last three years, the more honorary authors he or she reports, though the coefficients are equally small. Concerning differences across research fields, we find that scholars in economics, finance, and political science are about 40% less likely to report the presence of honorary authors in their last published paper than are members of the baseline group of interdisciplinary social scientists. By contrast, psychologists report even more instances of honorary authors in their published papers than interdisciplinary social scientists do.

Models 5 and 6 depict respondents' perceived assessments of the rates of ghost and honorary authors in their last published paper as the dependent variables. The results show that

higher the identified rates of ghost and honorary authors according to the ICMJE criteria are, the higher also the respondents' perceived the occurrences of these forms of authorship misconduct. Yet the effect for ghost authors is not significant at the conventional $p < 0.05$ level but only marginally significant at $p < 0.1$. Nevertheless, the results further attest to the discrepancy between the perception of authorship misconduct and its actual occurrence due to the relatively low explanatory power (as measured in Pseudo-$R^2$) of the regressions.

## Hypothetical assignments of authorship

The results on the prevalence and the perception of authorship misconduct revealed some preliminary evidence that the survey respondents seem to be applying an authorship definition that is fairly wide and does not adhere to a common point of reference such as the ICMJE definition. However, it is unclear whether the occurrences of these authorship issues derive from a lack of knowledge of common authorship criteria or from other considerations (whether conscious or unconscious) not mentioned in the survey questions.

To explore the two explanations further, we randomly assigned respondents into two groups and presented them with two vignettes. Table 2 depicts respondents' authorship assignments for these two hypothetical scenarios.

The first vignette lists the same scenario for both groups: A professor and a postdoc collaboratively write a paper and a student assistant supports the data collection. By design, the answers concerning who should be listed as an author between the two groups should not differ. The results indicate that the answers within the groups are nearly identical and the respondents similarly assign authorship. Almost all respondents (917 in Group A and 923 in Group B) award authorship to the professor and the postdoc, in concordance with the ICMJE authorship definition. Yet a substantial number of scholars (284 in the first group and 292 in the second group) deviate from the ICMJE criteria by awarding author credits to the student assistant who did not participate in the writing, revision, or submission process. This finding corroborates the prior conjecture that social scientists appear to have a broader conception of authorship.

Importantly, Models 1, 2, and 3 in Table 3 show that the authorship assessments do not differ significantly between the groups when we control for demographic and job-related factors, as the coefficient of *Group* is insignificant. This helps to establish a baseline behavior for the subsequent analysis when each group judges slightly different scenarios to explore the misconception of authorship criteria among social scientists in more detail.

**Table 2. Authorship assignments in the vignettes split by treatment group.**

|  | Group A | Group B |
|---|---|---|
| Total Assessments Vignette 1 | 973 | 984 |
| Professor Vignette 1 | 935 | 937 |
| Postdoc Vignette 1 | 955 | 968 |
| Student Assistant Vignette 1 | 284 | 292 |
| Total Assessments Vignette 2 | 975 | 986 |
| Postdoc/Professor Vignette 2 | 950 | 963 |
| Professor Vignette 2 | 607 | 700 |
| Student Assistant Vignette 2 | 82 | 87 |

The numbers correspond to the number of respondents who assigned authorship to the respective figure in the respective vignette. The number of observations differs between Vignette 1 and Vignette 2 because in each group two respondents in each group chose N/A options in the first vignette but not in the second vignette.

**Table 3. Regression results of hypothetical authorship assignments in the vignettes.**

|  | Vignette 1 | | | Vignette 2 | | |
|---|---|---|---|---|---|---|
|  | Prof. | Postd. | SA | Prof./Postd. | Prof. | SA |
| Group | -0.268 | 0.025 | 0.012 | -0.006 | 0.429 | 0.082 |
|  | (0.233) | (0.362) | (0.103) | (0.302) | (0.101) | (0.167) |
| Female | -0.014 | -0.411 | 0.010 | -0.133 | -0.060 | -0.193 |
|  | (0.247) | (0.390) | (0.113) | (0.321) | (0.110) | (0.189) |
| Anglophone | 1.081 | 2.603 | 0.044 | 0.709 | -0.195 | -0.017 |
|  | (0.482) | (0.623) | (0.225) | (0.499) | (0.232) | (0.352) |
| Continental Europe | 0.062 | 1.705 | 0.299 | 1.052 | -0.772 | 0.080 |
|  | (0.437) | (0.469) | (0.220) | (0.515) | (0.227) | (0.341) |
| Developing Countries | -0.350 | 1.051 | 0.631 | 0.145 | -0.503 | 0.513 |
|  | (0.460) | (0.509) | (0.243) | (0.540) | (0.255) | (0.363) |
| Age | -0.064 | -0.027 | 0.011 | 0.008 | 0.001 | 0.006 |
|  | (0.015) | (0.030) | (0.009) | (0.025) | (0.008) | (0.014) |
| Ph.D. Student | -0.320 | -1.036 | 0.117 | -0.713 | 0.260 | -0.267 |
|  | (0.405) | (0.609) | (0.199) | (0.482) | (0.196) | (0.359) |
| Professor | 0.136 | 0.197 | -0.181 | -0.118 | -0.194 | -0.198 |
|  | (0.294) | (0.499) | (0.135) | (0.412) | (0.134) | (0.214) |
| Editor | -0.945 | -0.597 | 0.070 | -0.337 | 0.094 | 0.370 |
|  | (0.265) | (0.414) | (0.129) | (0.366) | (0.130) | (0.196) |
| Years in Academia | 0.044 | 0.004 | 0.001 | -0.012 | 0.002 | 0.002 |
|  | (0.016) | (0.031) | (0.009) | (0.026) | (0.009) | (0.014) |
| Published Papers | -0.009 | -0.188 | 0.083 | -0.140 | 0.136 | 0.255 |
|  | (0.101) | (0.130) | (0.042) | (0.119) | (0.045) | (0.064) |
| Written Reviews | 0.319 | 0.036 | -0.037 | 0.236 | 0.004 | -0.169 |
|  | (0.109) | (0.133) | (0.039) | (0.128) | (0.038) | (0.065) |
| Business | 0.339 | 0.503 | -0.804 | -0.325 | -0.058 | -0.673 |
|  | (0.340) | (0.638) | (0.171) | (0.420) | (0.172) | (0.281) |
| Economics and Finance | 1.461 | 0.683 | -0.808 | 0.599 | -0.089 | -0.067 |
|  | (0.582) | (0.861) | (0.210) | (0.685) | (0.202) | (0.309) |
| Computer and Statistics | 0.813 | -0.830 | 0.345 | 0.732 | 0.642 | 0.595 |
|  | (0.436) | (0.564) | (0.183) | (0.618) | (0.208) | (0.266) |
| Political Sciences | -0.274 | -0.696 | -0.514 | -0.393 | -0.959 | -0.871 |
|  | (0.399) | (0.675) | (0.218) | (0.552) | (0.213) | (0.425) |
| Psychology | 0.791 | 0.762 | -0.637 | Perfect Predictor | 1.044 | -1.315 |
|  | (0.668) | (1.127) | (0.260) |  | (0.318) | (0.560) |
| Sociology | 0.076 | 0.894 | -0.225 | 0.107 | -0.506 | -0.088 |
|  | (0.435) | (1.120) | (0.221) | (0.687) | (0.221) | (0.352) |
| Chi-Square | 81.01 | 47.40 | 120.42 | 20.69 | 169.26 | 86.48 |
| P > Chi-Square | 0.000 | 0.000 | 0.000 | 0.241 | 0.000 | 0.000 |
| Pseudo R-squared | 0.117 | 0.142 | 0.051 | 0.048 | 0.069 | 0.076 |
| Observations | 1931 | 1931 | 1931 | 1935 | 1935 | 1935 |

Coefficients correspond to marginal effects derived from logistic regressions with standard errors in parentheses. The (omitted) reference categories for the categorical variables are: *Asia* for geographical region, *Postdoc* for job position and *General Social Sciences* for research field. S6 Table includes confidence intervals, p-values and more information on the varying number of observations.

In the second vignette, the assignment of authorship to the primary researcher (a postdoc for Group A and a professor for Group B), who takes over most of the tasks, differs only very slightly in comparison to the assignment of authorship in the first vignette. 950 respondents in Group A and 963 respondents in Group B assign authorship to the primary researcher. These assignments concur with the ICMJE authorship criteria. Yet respondents in both groups award authorship less often to the second researcher (a professor) who only performs the minimum number of tasks (revising the research design, revising the paper, and approving the submission of the paper) that would fulfill the ICMJE authorship criteria. In Group A, 607 respondents assign authorship to the professor. In Group B, 700 respondents assign authorship to the professor. Again, the responses deviate substantially from common conventions of authorship. More importantly, however, they even tend to withhold authorship. Differences between the groups remain highly significant even when controlling for various demographic and job-related factors. Model 5 in Table 3 shows that respondents in Group B were 42.9% more likely to award authorship to the second researcher.

Summing up, we find differences in the authorship assignment practices not only between the two vignettes but also across the two groups. The respondents less often assign authorship to the second-mentioned researcher if the primary researcher was a postdoc rather than a professor.

## Robustness of results

We conduct several robustness and endogeneity checks. First, while we are unfortunately not able to conduct a non-respondent analysis due to the preservation of respondents' anonymity, we analyzed the levels at which respondents disengaged from the survey. We compare the characteristics of those respondents that disengaged from the survey after filling out the insensitive questions on demographic, job and academic positions to the sample respondents. S8 and S9 Tables show the descriptive statistics as well as the results from a Wilcoxon rank sum test (we use this non-parametric test because there are 137 respondents who solely answered the insensitive questions). There exist only two significant differences: Those who only answered to the insensitive questions had on average more co-authors as well as more contributors on their last published paper. This points out that respondents most likely disengaged from the survey because the survey constituted too much work for them but not because of the existence of sensitive questions. On the page following the insensitive questions, respondents had to indicate all of their co-authors' and contributors' participation for each task. Consequently, the second stage of the survey was more extensive for participants with more co-authors and/or contributors, thus leading to them quitting more often. However, this deviation does not seem to heavily impact the composition of our sample because our sample average of 2.83 authors per paper (S3 Table) lies within the social scientific field averages [73].

First, we run all models with robust standard errors. The levels of significance remain invariant. Second, we estimate all models using OLS regressions and calculate variance inflation factors (VIFs) to assess whether our models suffer from multicollinearity. All VIFs are below the conservative threshold of 5 [74] and thus we have no reason to believe that our models suffer from multicollinearity. Third, we apply firthlogit, a special form of a logistic regression that considers rare events, to correct for the relatively low number of 64 ghost authorship observations [75]. The results remain invariant. Fourth, the average team size differs across research fields [76]. Therefore, we estimate the same regressions with the percentages of honorary and ghost authors in the supplementary material. The only difference is that psychologists no longer report a higher rate of honorary authors anymore. Fifth, we run all analyses including only respondents of papers with a maximum of five authors and contributors to

avoid any issues arising from the approximations of respondents stating that their last papers included more than five authors/contributors. The significance levels and implications do not differ from the findings above. Sixth, to detect potential unobserved heterogeneity within the groups, we create interaction terms with the product of *Group* and all exploratory variables for Models 4 and 5 in Table 3 and again run logistic regressions. The results are available in S7 Table. The coefficients of *Group* change only slightly, and the significance levels remain invariant. Last, Model 5 in Table 1 might suffer from sample selection, since honorary authorship increases the number of authors and lowers the number of contributors. The inclusion of only papers with at least one contributor in the analysis of the relationship between the perception and the occurrence of ghost authors may, therefore, suffer from endogeneity. To investigate this issue, we employ a Heckman two-step regression by applying *Honorary Authorship* as the selection variable. This regression returns an insignificant inverse Mills ratio. Therefore, our results do not suffer from sample selection bias [77].

## Discussion

We analyzed the prevalence of ghost and honorary authorship in the social sciences. We find that many researchers apply very broad authorship criteria that do not accord with the criteria laid out by the ICMJE. This is remarkable, as many social scientific societies subscribe to institutional arrangements such as COPE (which attributes authorship based on ICMJE guidelines) [41]. Nevertheless, the results are in line with prior work that indicated deviations from the ICMJE standards within the life sciences [19,46]. Interestingly, the social scientists in our study report more honorary authorship but less ghost authorship than life scientists [49,78], using ICMJE criteria as the point of reference.

We also investigated how the misattribution of authorship comes about. By and large, our results show that researchers tend to award authorship more broadly to junior scholars and at the same time may withhold authorship from senior scholars if those are engaged in collaborations with junior scholars. Many social scientists in the sample we studied believe that more of their non-author collaborators should receive authorship despite the fact that many of them do not fulfill the ICMJE standards. We thus find a general pattern that scholars tend to be more generous when it comes to assigning authorship. This may imply that social scientists have their own authorship criteria in mind that do not necessarily match commonly applied criteria such as those laid out by ICMJE. Misattributions of authorship can go both ways: the results from the second vignette show that a substantial number of respondents are more restrictive and even tend to withhold authorship from senior researchers who work with junior scholars. As noted above, in the second vignette, we presented a scenario where both scholars should receive authorship following the ICMJE criteria, despite unequal inputs.

In sum, our vignettes document that the prevalence of ghost and honorary authorship is to a large extent affected by the fact that many participants did not adhere to common authorship criteria. Even more so, the discrepancy in authorship attribution could be exacerbated by fairness expectations and benevolent discrimination.

We show that some social scientists withhold authorship from individuals if they collaborate with others who put more effort into a research project. Fairness expectations provide a plausible explanation for this finding: Scholars perceive it as unfair to award authorship to all researchers if the distribution of efforts is highly uneven. After all, even the best social scientists are still human beings and thus learn fairness expectations and altruism from early childhood on [79].

Benevolent discrimination "is a subtle and structural form of discrimination that is difficult to see for those performing it, because it frames their action as positive, in solidarity with the

(inferior) other who is helped, and within a hierarchical order that is taken for granted" [80]. Following this reasoning, it appears that researchers in our sample award authorship more generously to undeserving student assistants and withhold authorship from professors if they collaborate with junior scholars without sharing the work equally.

Moreover, the results from the empirical study exhibit suggestive evidence that benevolent discrimination appears not only in the hypothetical assignments but also in actual authorship assignments. Ph.D. students are much more likely than faculty members to have encountered cases of honorary authorship. Even the ethics policies from large social scientific societies like the Academy of Management [29], the American Sociological Association [30], and the American Psychological Association [31] enforce benevolent discrimination by requiring Ph.D. students to become the first author when published articles are based on their dissertation without considering the distributions of the contributions in such research projects.

As it concerns the determinants of authorship misattributions, our results report a clear gender difference. Women are more likely to report that honorary authors were included in their papers. While our results do not speak in favor of discriminatory effects here, they highlight an area for future inquiries to study gender inequalities in the social sciences [66].

Hierarchical pressure might explain some regional differences in the prevalence of ghost and honorary authorship. Scholars living in Anglophone and Central European countries report that authorship misconduct is less prevalent in their research. Work by researchers residing in Asia, by contrast, more often includes ghost and honorary authors. The cultural background of these scientists coming from countries with generally higher levels of power distance [81] might explain this phenomenon, as department or faculty heads may receive authorship even without having read the paper [67,82].

Our research also adds an observation to the literature on highly prolific authors. Awarding authorship to department or faculty heads might also explain why respondents who had published more papers also more often had honorary authors in their papers and assigned honorary authorship more often in the vignettes. More generous authorship assignments may result in higher publication counts [65,83].

Last, we show that the prevalence of ghost and honorary authorship varies across research fields. Different author ordering conventions might explain this. Economics, finance, and political science usually rank authors alphabetically [84,85] while other fields, such as psychology, order by contribution. Consequently, in the former fields, awarding authorship to yet another author means that the main author(s) may disappear into the "et al." rubric for future citations. For the latter fields, there are more incentives to include individuals in charge of financing the research project (e.g. the department head) as the last author, for example, even if they had not participated in the research process [86]. Arguably, ordering by contributions rather than alphabetically increases the chances of honorary authorship assignments.

## Implications

Our results highlight a high prevalence of ghost and honorary authorship and a broad deviation in authorship assignments from such as the commonly employed ICMJE criteria. These findings highlight the necessity to introduce authorship criteria that are better tailored to the needs, preferences, and perceptions of social scientists. Hence, we call upon large research societies such as the Academy of Management, the American Economic Association, the American Psychological Association, and the American Sociological Association, as well as on the most prominent publishers such as Elsevier, Taylor & Francis, and Springer [23], to revise their existing guidelines on authorship. The revision should focus on establishing clear, precise, and specific criteria that should be also discussed and taught at their annual meetings.

This would increase knowledge and awareness of authorship and would ease the job of journal editors as they could rely on accurate guidelines when making attributions of authorship. In turn, these attributions would lead to fairer comparisons of authors for career decisions. Though, one might account for more multi-dimensional criteria in these comparisons anyhow. As a case in point, Moher et al. [87] highlighted several methods of assessing scientific performance including creativity, openness, transparency, and addressing and solving societal problems that go beyond pure publication counts.

The differences in authorship attributions in our study highlight the need to rethink contribution disclosures. Contribution disclosures allow insights into the workload distributions among author teams. Even if researchers' understanding of authorship varies, contribution disclosures would give an outside observer a better chance to assess who did what regarding a specific publication. This would not rule out misconduct, of course, but it would make it much harder to add individuals who did not contribute at all. If journals require contribution statements from each author and contributor, scholars will face high coordination efforts to submit factually wrong but congruent contribution disclosure statements. Hence, mandatory contribution disclosure statements raise the barrier for submitting falsified author lists. The introduction of contribution disclosures has already reduced instances of honorary authorship in the life sciences [33]. Moreover, contribution disclosures might also reduce instances of benevolent discrimination, as uneven divisions of efforts are accounted for in the contribution statements.

Of course, the introduction of contribution disclosures is not a panacea. It requires editors and authors to be aware of authorship criteria and the consequences of potential misconduct. To ensure this, we recommend that academic societies, research institutions, and publishers establish social scientific authorship criteria in their guidelines and provide tutorials and workshops on authorship.

## Limitations and future research

Every research has to be understood and interpreted in light of its limitations. Perhaps the greatest limitation of this study is that existing authorship criteria in the social sciences are overly vague, leaving researchers with very little guidance. In our work, we applied the ICMJE authorship criteria as a common point of reference to better understand authorship assignments in the social sciences. Although universities and societies should apply these criteria (through associating with COPE, for example), social scientists' perceptions about what constitutes authorship clearly differ from the ICMJE criteria. We attribute this among others to inherent differences in the research and authorship attribution processes of social and life sciences [21]. Consequently, though we find that ghost and honorary authorship are common, some social sciences researchers might argue that this does not represent misconduct but rather is reflective of common authorship practices in the social sciences.

Also, the results of our study may not necessarily generalize. Our findings reported a relatively low number of identified ghost authors. While the application of firthlogit indicates that the findings are robust and valid, the small number of observations might be obfuscating further effects. The quantitative survey contained sensitive questions that, despite the fact that respondents were assured of anonymity, could lead to the understatement of actual wrongdoings–a common problem in research on questionable research practices and academic misconduct [68]. For this reason, the actual prevalence of ghost and honorary might be higher than what our study uncovered, especially when using survey designs that explicitly deal with sensitive items.

Also, the unequal distribution of researchers among geographical regions and research fields might reduce the applicability of our results. Some fields might exhibit higher or lower

levels of ghost and honorary authorship. More field-specific studies could determine the prevalence more precisely.

Several fruitful areas for future research can be derived from this study. First, future research could discuss whether common authorship practices in the social sciences like automatically assigning Ph.D. students first authorship should be continued or replaced by more merit-based mechanisms. Second, comparing research-field-specific authorship criteria could explain why scholars from different disciplines vary in their authorship assessments. Third, closely examining the authorship assignments of extremely prolific social science scholars could clear up doubts on whether they really exhibit higher productivity levels or receive honorary authorship more often. Fourth, a qualitative study surveying researchers who experienced ghost and/or honorary authorship could provide a better understanding of the motivations and consequences of authorship misattribution. Last, the use of anonymity-preserving survey measures such as item-sum techniques [88] could increase respondents' perceived anonymity and therefore lead to a more accurate assessment of ghost and honorary authorship.

## Conclusion

This study analyzed the prevalence of ghost and honorary authorship in the social sciences. Our results show that social scientists perceive authorship differently than established in the ICMJE criteria and allied organizations that seek to create standard definitions. We find that authorship misconduct, in the form of ghost and honorary authorship, is highly prevalent in the social sciences. We also investigated the correlates of authorship misassignments and found that fairness expectations and benevolent discrimination are prime candidates to explain why and to what extent researchers may either assign authorship too freely or restrict it too much. We discuss potential solutions: The introduction of social scientific authorship criteria by the largest research society, the enforcement of contribution disclosures through journals and publishers, and a shift away from the importance of citation and publication counts in hiring and tenure processes could all alleviate the problem of misattributing authorship and distorting publication records.

## Supporting information

**S1 Table. Descriptive statistics for dichotomous variables.**
(PDF)

**S2 Table. Descriptive statistics for integer variables.**
(PDF)

**S3 Table. Pairwise correlations coefficients for all employed variables.**
(PDF)

**S4 Table. Regression results of actual and perceived prevalence of ghost and honorary authorship including confidence intervals.**
(PDF)

**S5 Table. Regression results of hypothetical authorship assignments in the vignettes including confidence intervals.**
(PDF)

**S6 Table. Regression results of hypothetical authorship assignments in the vignettes.**
(PDF)

**S7 Table. Comparison of samples for dummy variables.**
(PDF)

**S8 Table. Comparison of samples for integer variables.**
(PDF)

**S9 Table. List of journals and societies included in the data collection.**
(PDF)

**S1 File. Do-file with Stata code for generating all figures and tables.**
(DO)

**S1 Text. The text of the vignettes employed in the study.**
(PDF)

## Acknowledgments

We thank the members of the Department of Business Decisions and Analytics of the University of Vienna, especially Oliver Fabel and Rudolf Vetschera, for their inputs on this paper.

## Author Contributions

**Conceptualization:** Gernot Pruschak, Christian Hopp.

**Data curation:** Gernot Pruschak.

**Formal analysis:** Gernot Pruschak.

**Investigation:** Gernot Pruschak.

**Methodology:** Gernot Pruschak, Christian Hopp.

**Project administration:** Gernot Pruschak.

**Resources:** Gernot Pruschak, Christian Hopp.

**Software:** Gernot Pruschak, Christian Hopp.

**Supervision:** Christian Hopp.

**Validation:** Gernot Pruschak, Christian Hopp.

**Visualization:** Gernot Pruschak.

**Writing – original draft:** Gernot Pruschak.

**Writing – review & editing:** Gernot Pruschak, Christian Hopp.

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
