## [Decision Letter · Decision Letter 0]

12 Mar 2021

PONE-D-20-30365

And the credit goes to … - Ghost and honorary authorship among social scientists

PLOS ONE

Dear Dr. Pruschak,

Thank you for submitting your manuscript to PLOS ONE. After careful consideration, we feel that it has merit but does not fully meet PLOS ONE’s publication criteria as it currently stands. Therefore, we invite you to submit a revised version of the manuscript that carefully and systematically addresses all the points raised by the three reviewers during the review process.

All three reviewers explain clearly the revisions that they consider necessary for your manuscript to reach publishable standards. Reviewers 1 and 3 also provide detailed comments and suggestions pertaining to specific parts of the text of your manuscript. It is important that you address all of these, explaining specially in instances where your responses may not be exactly the same as the reviewers suggest you need to do.

We look forward to receiving your revised manuscript.

Kind regards,

Emmanuel Manalo, PhD

Academic Editor

PLOS ONE

Journal Requirements:

4. Please note that the ICMJE statement now lists a fourth authorship criterion, namely "Agreement to be accountable for all aspects of the work in ensuring that questions related to the accuracy or integrity of any part of the work are appropriately investigated and resolved." http://www.icmje.org/recommendations/browse/roles-and-responsibilities/defining-the-role-of-authors-and-contributors.html#two.

5. Please improving statistical reporting and refer to p-values as "p<.001" instead of "p=.000". Our statistical reporting guidelines are available at https://journals.plos.org/plosone/s/submission-guidelines#loc-statistical-reporting.

6. In your methods section, please provide additional information about your participant inclusion and exclusion criteria as well as the selection criteria for the journals from which you selected authors. Please also provide the list of journals as supporting information.

Reviewers' comments:

Reviewer's Responses to Questions

**Comments to the Author**

1. Is the manuscript technically sound, and do the data support the conclusions?

Reviewer #1: Partly

Reviewer #2: Yes

Reviewer #3: Yes

2. Has the statistical analysis been performed appropriately and rigorously? 

Reviewer #1: I Don't Know

Reviewer #2: Yes

Reviewer #3: Yes

3. Have the authors made all data underlying the findings in their manuscript fully available?

Reviewer #1: Yes

Reviewer #2: No

Reviewer #3: No

4. Is the manuscript presented in an intelligible fashion and written in standard English?

Reviewer #1: No

Reviewer #2: Yes

Reviewer #3: Yes

5. Review Comments to the Author

Reviewer #1: PONE-D-20-30365

This paper presents a survey study on gift and ghost authorship in the social sciences. Before speaking to the content of the paper there is the research ethics review situation. According to the authors, this study did not obtain approval for the survey study. The authors explain that social science studies do not requires IRB approval in Germany. Does PLOS allow this justification? If so, I suggest that authors could make explicit the policies (regulations) or institutional norms that protect human subjects (e.g. federal data protection, confidentiality mechanisms, informed consent).

Given the limited literature regarding authorship in this field, this paper does present some interesting empirical findings regarding rates of honorary and ghost authorship a as well as power and gender. Although I do believe that such results should eventually be published, the paper still has important shortcomings at this point.

Major issues:

1. There is a generalized vagueness throughout the paper which makes it difficult for the reader to follow. For example, it is unclear how multiple authors create better quality and higher productivity papers. It is also unclear why the simple presence of multiple authors gives way to malpractice. There was questionable practices before multiple authorship. This is further developed in the specific issues section below. A thorough review of the English may also help.

2. The authors often state that there is no applicable authorship criteria in social sciences (e.g.p.30) However, there is authorship guidance in various professional societies (e.g. American Sociological Association Code of Ethics and the American Psychological Association Code of Conduct). Moreover, the Committee of publication ethics does have some guidance on authorship; https://publicationethics.org/authorship. Lastly, certain journals, research centers as well as funding agencies have developed some guidance that apply to social scientists

3. Surprisingly, even though the author says there is no criteria, the authors seem to explain that there is a correct and incorrect way to distribute authorship. Even in the abstract, the authors seem to want to evaluate if scholars assign authorship “correctly” in three hypothetical situations. However, if there is no criteria how could there be a correct and incorrect way to distribute authorship.

4. The recommendations at the end of the paper (p.27) do not logically follow from the actual results of the survey. For example, the author mention the importance of contributor declarations but never explains why this would be important. If a person lies about authorship they could also simply lie about contributorship. Also, ghost authors will not be listed as contributors. The authors also mention that there are power dynamics and thus whistleblowing should be used to go outside the system and change past submissions. Why is whistleblowing used to counter power-dynamics as opposed to other methods? Moreover, changing the past submission would have a huge impact on science. Not only would authors change on past papers, so would the citations and refences. Do we also take tenure away from people after the fact? Feasibility and cost would be a serious consideration. This is almost a completely different paper and argument.

Minor comments:

In many countries, the notion of malpractice is often a legal term. Is this term found in Germany in policy? To my knowledge authorship regulations are not found directly in legal documents. It may be lack of compliance, misbehavior or questionable research conduct.

p.3 l. 42 There are statements that are often vague and not explained. For example, the first sentence “Publish or Perish” epitomizes the academic reward system across scientific disciplines nowadays better than ever before”. Why is this notion more important now as compared to before? When did this shift happen?

p.54-56; sentence non-sensical. I am not sure what is the gold standard in this sentence. Often we say peer-review is the gold standard. I have not heard this about citation and publication counts being the gold standard.

p.4 l.61 As the authors mention, there is a lot more literature in life sciences and medical science when compared to social sciences. However, the author should also cite the literature that does exist regarding the rise of co-authorship in social science (1). The author can also look at studies on interdisciplinary that include a cross-disciplinary lenses(2–8).

p. 4 l. 63 author suggest that contributor declarations have not been included in social science journals. There are no citations and it is not a commonly accepted fact. There are journals of an interdisciplinary nature that include social science work (e.g. PLOS). Some social science journals may be gathering the information about contributions during paper submissions without actually publishing the content. Also, if there are no citations to illustrate this fact. It would be relevant to exemplify this point by citing journal policies.

p.6 l. 103 ; please explain Harbring and Irlenbush study. This is not clear.

p.10 l. 183; I am unsure what the antepenultimate and penultimate task would be. What is the “last task”. Is this the one that takes the least amount of time? The most value? The idea? The methods? The manuscript writing?

p.11 203-205 ; unclear

l. 206=209 ; what is collaboration on eye level?

l.213 – workload shift should be division of labor.

p.25 l.444-447. I am unsure what result leads to the notion that not everyone knows what constitutes authorship. There may simply be differences in interpretation without the individuals not understanding the concept.

p.25 l.448-452. The author mentions objective authorship criteria. Although there are regulations in which individuals must have substantially contributed to the research to be included as author. The notion of “substantial contribution” is an ongoing debate. Thus it remains a subjective notion. If the author believes there is an objective notion at play, please explain.

p.26. Although Weinstein may have had an impact on this context, there is also increased awareness regarding gender inequities in science. The author should explain the negative impact of this inequities by referencing the literature on the subject.

p.27 l.475; the authors state that researchers from Asian countries have stronger likelihood of obedience. As written it is construed as stereotyping. Can you refer to the literature to explain the hierarchy in certain groups?

p.26. l. 444 “This accords to the authorship guidelines of their academic societies: Articles based on dissertations should always include the PhD students as authors” Can you cite these academic societies. Is this also the case in university regulations?

p. 26 I am not sure what the author considers as elderly authors. Is this actually based on age? Is is based on academic seniority? The notion “elderly” has a very negative connotation and should be avoided.

p. 26. When talking about gender please use the notion women/men. Gender is a social construct. Avoid the notion of male/female that refers to sex as a biological variable.

p. 29 the authors mention a difference between authorship and effort which is certainly true. They conclude that because there is a difference, research institutions should rely on other estimates. What are other estimates and why and how would one include the notion of effort in evaluation?

p.31l. 554 I am unsure what expressiveness implies in this context.

Methods and Results: please ensure that a second peer-review looks over statistics. I am not a quantitative researcher.

Reviewer #2: This paper reports results of a survey of authors of social sciences articles showing that ghost and honorary authorships occur in the social sciences, and correspondingly suggests that authorship standards should be implemented in the social sciences, much as they already are in certain journals in the hard sciences.

Overall, the paper is well done.

The one issue I see is with vignette #3. As written, this vignette mingles the authorship issues with a severe ethical issue. In this vignette “After the successful graduation of the PhD student the professor creates a scientific paper based on the empirical results of the thesis and submits it to a journal.” The vignette does not state if the masters student was listed as an author. It does not state if the masters student gave permission for the professor to turn the thesis into a paper. While the professor provided input to the study and thesis, the vignette makes it look like the professor is appropriating the work of the student. If this is indeed the case, then I have a hard time trusting the results of the survey for this vignette because it mingles the ethical issue (appropriation of work) with the authorship issues. I would imagine that many of those completing the survey would have had the same reaction, and that their answers regarding authorship would have been clouded by the ethical issue.

Given the potential confounding of vignette #3, I would suggest it be removed from the paper unless the authors can make a convincing case that no confounding occurred. In my view, removal of this vignette would not really weaken the conclusions. If the results of this vignette are removed, it would still be worth keeping something about it in the paper to highlight the ethical issue.

Regarding data availability, I would suggest that the authors add a table (or supplement) with the list of journals/conferences associated with the papers/authors who answered the survey. The list of sources is important and cannot be considered confidential. Also, I would suggest the authors do what is necessary to fully anonymize the survey data and to make it available. Although they say “available upon request (with caveats)”, in practice this tends to simply delay the availability question and in most cases authors then decline to share data. I think data availability must be addressed before the paper is published.

Reviewer #3: In this study, the authors conduct a survey of publishing authors in the socials circles in o order to understand prevalence of ghost and honorary authorship in the social sciences. Ghost authorship is when a contributor fits the definition of an author, according to the ICMJE, but wasn’t an author on a paper. Honorary authorship, in contrast, is when the contributor does not meet the criteria of an author, but is an author on the paper anyway. The authors of this study ask respondents about their\\ most recent paper, and also have respondents assign authorship to authors in three vignettes. Using these results, the authors conduct an exploratory analysis of the factors relating to actual and perceived ghost/honorary authorship, and also at how surveyed authors perceive contributions in the vignettes. The findings are that ghost/honorary authorship is a common issue in the social sciences, yet given their responses to the vignettes, authors also have a strong working definition of what constitutes authorship. The authors interpret this to mean that, at least in the social sciences, perceptions do not match reality when it comes to authorship criteria, and some people are undeservedly being added or left off of authorship bylines.

The topic is interesting from a scientoemtrics and science evaluation standpoint, and should have interest to general readers, though the authors could do a better job motivating their study. The quality of the writing was variable throughout, and I would appreciate work improving the clarity of the writing, as well as structuring the manuscript in a more readable way. The methods and statistical analyses are fair, although I would appreciate more detail on both. My largest issues are with the framing of the paper and the findings, specifically with the choice of the ICMJE as the “correct” criteria, and the decision to frame the interpretation of results in terms of ‘correct’ and ‘incorrect’ responses.

Below I provide some major and minor comments which I hope will improve the quality of the manuscript.

# Major comments:

Major Comment 1:

My biggest issue with the manuscript is the use of the “correct/incorrect” lens for interpreting results. Specifically, I think that this sentence in the discussion is problematic, as well as similar text throughout: ‘In addition, social scientists’ perception of the occurrence of authorship malpractices provide grounds to suspect that not everyone might be aware of what constitutes authorship.”

But perceptions are important! What constitutes authorship on a paper is really a norm of a disciplinary field, and so it would make sense that the norms of medical and life sciences (exemplified in the ICMJE criteria) would not transfer cleanly to the social sciences, just as we wouldn’t believe that the norms of High-energy physics to transfer cleanly to the life sciences. Ideally, the authors would use a more Social science-based criteria for determining the “correct” contributions, however no such criteria exists. But if a criteria were created, I imagine that it would be different than the ICMJE, and based on social scientist’s perceptions of contribution in their own field.

My point here is that a “significant contribution” to the typical social scientist will likely mean something very different than to the typical life scientist. Therefore, using a “correct/incorrect” framing, based on the ICMJE, is very strongly applying one field’s standard onto another, where it perhaps isn’t warranted.

I suggest that the authors take a more cautious approach to interpreting the results, particularly avoiding language of “correctness/incorrectness”, and careful not to dismiss the importance of social scientist’s perceptions. The use of ICMJE is mostly warranted, if imperfect(see Major Comment 2), but its imperfection should be a factor in interpreting results.

Major comment 2:

Norms of authorship and contribution vary according to field. What warrants a co-authorship in one field, like life sciences, may warrant only an acknowledgement in another. In a particularly-interesting case, students of the physicist Stephen Hawking received no credit at all! (Though the prestige of their posting still granted them great career success).

In this paper, the authors use a definition of contribution from the ICMJE, an organization in the life and medical sciences, and apply it to situations in the social sciences. Because notions of contribution can vary so widely between fields, the choice of the ICMJE definition needs to be better motivated, and the potential consequences discussed.

One way to better motivate the choice of ICMJE is to make it explicit that PLoS also bases their definition of contribution based on the same guidelines, and PLoS, as you are aware, published a great deal in the social sciences.

The paper would also benefit from discussions of other definitions of contribution, such as those used by Nature journals. For example, would an alternative definition of contribution change the “correct” answer of any of the vignettes?

The pages with PLoS and Nature authorship guidelines are linked below:

https://journals.plos.org/plosone/s/authorship

https://www.nature.com/nature-research/editorial-policies/authorship

Major comment 3:

Aspects of the study are introduced in the methods section without being motivated in the introduction. For example, the authors examine gender, country, age, and more in relation to ghost authorship, however the introduction lacks any mention of any demographic items and their significance to ghost and honorary authorship.

I suggest the authors flesh out the introduction somewhat, bringing up the importance of professional rank (postdoc vs. professor) and gender in order to set the narrative for exploring these variables later in the analysis.

The discussion on these variables is also quite light, and is not well-contextualized in the existing literature. I would appreciate a more in-depth and referenced discussion that considers how gender and other demographics relate to authorships. For example, on the gendered division of labor in science, and also the “Matilda Effect” in science. I linked both of these below.

https://doi.org/10.1177/0306312716650046

https://www.jstor.org/stable/285482

Major Comment 4:

Assuming I’ understanding the authors correctly, I don’t believe the the author’s interpretation of the Central Limit Theorem is correct (in section Methods.C). The CLT has to do with repeated samples from an underlying distribution, such that taking the mean of repeated samples (say, samples of size 100 from a population of size 10,000) will results in those means being normally distributed. Here, the authors take a single sample of 2,000 or so survey responses from some underlying population of scholars. As such, the CLT, at least in my understanding, doesn’t seem to make sense here. The authors should update Methods.C accordingly.

I would also feel more comfortable with additional diagnostics to determine whether the models fit statistical assumptions. One issue, for example, could be multicollinearity, or the degree to which independent variables are correlated, which can cause problems in logistic regression. There should be a function in STATA or some other statistical software (the `vis` package in R, for example) to run a test on this assumption.

Additionally, “# of ghost authors”, “# of honorary authors”, “perceived ghost authors”, and “perceived honorary authors” all appear non-normally distributed, and even zero-inflated in figure figure 2. This might confound the results from linear regressions using these values. I wold advise adding either a) diagnostics of the model, such as Q-Q plots, to demonstrate that model assumptions are not violated, or b) additional robustness checks to ensure that main results don’t qualitatively change under different models. For example, a zero-inflated Poisson model on “# of ghost authors”.

Major Comment 5:

The authors use t-tests and linear regression coefficients throughout, but there is little discussion of the effect size, or in other words, how meaningful or trivial the observed effects are. For example, the authors state in line 292 that “that the perceived share of honorary authors is significantly lower than

the identified share of honorary authors (t=7.5946; df=1841; p=0.0000).”, but how meaningful is the exact difference?

I advise that the author include a mention of the effect size when discussing these statistical tests. For example, in the above sentence, include the difference in the shares of perceived and actual honorary authors. Similarly, the authors should contextualize the coefficients from the regression, discussing how much of the variance they actually explain in the analysis.

This also brings me to another point—that the R2 values, or variance explained by these regressions are all very small. This isn’t a problem, it is exploratory after all, but consider incorporating this fact into the paper’s discussion, that even the strongest coefficients explain only a small part of the variance in the data.

# Minor comments:

- The authors find that “authorship issues also occur in the social sciences”, but is there any evidence that it is better/worse than in the life sciences? Other fields? Also, is there any evidence that policies implemented to combat these issues, such as contribution disclosures, have improved the situation in other fields?

- Line 28: This is a strong claim. Surely there is one social science journal that requires contribution disclosures? Or perhaps a Mega Journal like PLoS or Nature that requires them, and also published them in the social sciences.

Line 46: This sentence implies a causal mechanism for collaboration, namely, that researchers do it to bolster their productivity. However, I'm not confident that this claim is supported by the cited papers. For example, people might collaborate because information technology makes it easier, or because modern scientific research is highly-complex, and productivity is not a consideration. I advise being a little more cautious with the wording here.

- Line 109: I’m not sure what “aspire” means here, replace “might not aspire” with “might decline”

- Line 110-112: a reference should be given here for this claim about freelancers

- Fig 1: pie charts are difficult to read and interpret. Replace figure 1 with a bar chart.

- The terms “ghost” and “honorary” authorship are used in the abstract and introduction, but not formally defined until the literature review. A brief plain-language definition should be given for each of these terms at their initial use, in both the abstract and intro.

- A helpful paper for section C of the intro, in relation to honorary authorships, could be the “Chaperone effect in scientific publishing”: https://www.pnas.org/content/pnas/115/50/12603.full.pdf

- Line 159-160: The 20% and 50% numbered quoted appear to come from an article cited in 1993 (I don’t have access to it). This is an old reference, and surely not representative of contemporary acceptance rates. I suggest finding an updated reference and set of numbers. A starting point could be this paper: https://www.sciencedirect.com/science/article/abs/pii/S1751157713000710

- The authors first state that they used “Qualtrics” on line 200, but this should be brought up much earlier, at the beginning of the methods. Really, Methods.B should be made the first section of the methods.

- line 206: I’m not sure what “eye-level” mean there, perhaps another term could be used?

- line 235: the authors state that they collect authors from “well renowned expert journals”, but how were these journals and conferences selected? Are they representative of the social sciences as a whole, or are they skewed towards certain fields?

- line 244: “ballot stuffing” has a very specific, political meaning in the U.S. at least, and so I would rephrase this sentence to avoid it.

- Line 456: Contrarily -> Contrary

- Line 465: the authors should give a few more words about the Weinsten Scandal, as some readers may not immediately know what it is in reference to.

- Line 530: Is there evidence that hiring committees are not already screening applicants along a variety of criteria? #Publications, while important, rarely sees like the -only- criteria used in hiring and promotion decisions.

6. PLOS authors have the option to publish the peer review history of their article (what does this mean?). If published, this will include your full peer review and any attached files.

Reviewer #1: **Yes: **Elise Smith

Reviewer #2: No

Reviewer #3: No

---

## [Author Response · Author response to Decision Letter 0]

27 Sep 2021

Please refer to the attached responses to the reviewers for our detailed responses to the reviewers' valuable comments.

---

## [Decision Letter · Decision Letter 1]

28 Feb 2022

PONE-D-20-30365R1And the credit goes to … - Ghost and honorary authorship among social scientistsPLOS ONE

Dear Dr. Pruschak,

Thank you for submitting your manuscript to PLOS ONE. After careful consideration, we feel that it has merit but does not fully meet PLOS ONE’s publication criteria as it currently stands. Therefore, we invite you to submit a revised version of the manuscript that addresses the points raised during the review process.

I was appointed as academic editor of the paper on 20 January 2022. I read the long editorial history of the manuscript and the two reviewers' reports agreeing on minor revision. I apologize in advance for highlighting, only at this stage, a couple of points that requires careful revision.

A first very general point is about the design of the sample for the survey. You started from a sample of 63,240 scholars (mail addresses). Due to errors in mails the sample was reduced to 47,697 recipients. Respondents were 2,817.  The respondents represent 4.5% of the sample (5,91% of the recipients). The final number of completed surveys used for analysis are 2,222, i.e. 3.5% of the sample. This point should be also indicated

This dramatic loss of respondents should be discussed more carefully than in the current version of the paper, also in reference to the presence of sensitive questions in the survey. It appears that the authors do not design the survey for handling the presence of sensitive questions and the possible self-selection of respondents. Hence, results should not be generalized to ‘social scientists’ such as in the abstract rr. 29-30 and in the conclusion rr. 689-690. More in general, I think that the paper should be more cautious about the general robustness of the survey.

A second point is about the regression analysis. From table S5 and S6 it appears that confidence intervals for the big part of the considered carriable include zero, i.e. the estimates are not statistically significant. This is not highlighted in the presentation of the results. In particular, the discussion of gender effect is based on data that are not statistically significant; but this is true also for other variables. I would like to suggest a more careful interpretation of regression results and, possibly, the clear indication of significance level directly in Table 1 and 3.

We look forward to receiving your revised manuscript.

Kind regards,

Alberto Baccini, Ph.D.

Academic Editor

PLOS ONE

Reviewers' comments:

Reviewer's Responses to Questions

**Comments to the Author**

1. If the authors have adequately addressed your comments raised in a previous round of review and you feel that this manuscript is now acceptable for publication, you may indicate that here to bypass the “Comments to the Author” section, enter your conflict of interest statement in the “Confidential to Editor” section, and submit your "Accept" recommendation.

Reviewer #2: (No Response)

Reviewer #3: All comments have been addressed

2. Is the manuscript technically sound, and do the data support the conclusions?

Reviewer #2: Yes

Reviewer #3: Yes

3. Has the statistical analysis been performed appropriately and rigorously? 

Reviewer #2: I Don't Know

Reviewer #3: Yes

4. Have the authors made all data underlying the findings in their manuscript fully available?

Reviewer #2: No

Reviewer #3: Yes

5. Is the manuscript presented in an intelligible fashion and written in standard English?

Reviewer #2: Yes

Reviewer #3: Yes

6. Review Comments to the Author

Reviewer #2: Most of my comments have been addressed. However, my request that the journal/conference list be published has been ignored. The authors have successfully argued that the data cannot be made publicly available due to the potential for reconstruction of PII. I'll acquiesce on that. However, they cannot argue that the journal list should not be published. No PII can be inferred from a simple journal list without any other data. Addition of the list will add credibility to the paper in that readers will see names of journals they know.

Reviewer #3: I thank and commend the authors for the hard work of revising this manuscript. Its quality is greatly improved, and I especially appreciate how the ICMJE guidelines are re-contextualized to be less absolutist. In total, I feel that my comments and suggestions were adequately addressed, though I still have some minor issues which I believe should be addressed before submission, and which I detail below,

- The number of survey respondents is inconsistent. The abstract says 2,222, but the paper reports 2,223 in another case. Moreover, smaller numbers are used in the actual statistics analyses. I advise the authors to ensure consistency between these numbers, and make clear why numbers change throughout the manuscript. Moreover, the abstract should list the actual number of respondents used in the analysis, not the number before exclusions, so as not to overstate the size of the data being analyzed.

- Line 50: The use of “development of technologies” here is awkward, largely because a) technology is not listed in the corresponding reference, b) it is of a different kind to the other items in the list, and c) it is unclear whether “complex nature…” refers to “development of technologies”. Overall, I recommend cutting this phrase.

- Line 51: When mentioning the increasing complexity of science, the authors should cite the canonical paper in this area about the Burden of Knowledge: https://academic.oup.com/restud/article-abstract/76/1/283/1577537

- Line 70: The use of “falsify” is, while accurate, also very strong and brings to mind fraud, which I believe the authors don’t want to insinuate. I suggest replacing this with a work like “distort” or “confound” which has a similar meaning but with a less harsh connotation.

- Line 71: There is a double-period here, which should be fixed

- Line 81: the use of the phrase “and even” in the sentence “…methodology, software, **and even** writing, reviewing, and editing* is strange and may confuse the reader—I suggest simply cutting this phrase.

- Line 113: It is unclear when reading the list of ICMJE criteria whether an author must meet one of these criteria or all of them. The ICMJE website makes it clear that the author must meet -all- criteria by appending an “AND” at the end of each list item. I suggest the authors do the same, such that the list looks like: “…acquisition of data, or analysis and interpretation of data; and 2) drafting the article…”

- Line 137, in the sentence “First, pressure from co-authors, whose citation and publications counts usually benefit from a lower number of contributors can lead to researchers declining or being declined authorship;…”; while I see how credit itself (something that is abstract, social) is affected by adding co-authors, it is unclear how adding co-authors would affect the citation and publication counts of a paper, which in nearly all cases use “full-counting” to count every contribution equally. I suggest rephrasing this sentence to be more clear, or to provide a reference for how publication/citation counts would be affected.

- Line 165: I appreciate the authors adding in the chaperone effect, but I think the current description is a little misleading. The Chaperone Effect is not really about repetitional bias in peer review at elite journals, as the authors claim here. Rather, it refers to a kind of mentorship, where a senior author guides their student in how to write for a specific, elite, journal. If the authors would like a reference about the bias towards famous authors, I recommend this by Tomkins, Zhang, & Heaviln (2017): https://www.pnas.org/content/114/48/12708

- Line 203: I appreciate that the authors added a mention of the Matilda Effect which I mentioned, but I think that the description provided misses a key nuance—it is not only that women receive less credit for their work, but that the credit tends to go to male colleagues involved in the study. This can be potentially damaging in the case of honorary authorship, in which adding honorary senior (often male) colleagues can “steal credit” away form women in the team.

- Line 581: The authors should be more specific with “Anglophone”, “Central European”, and “Asian” scholars. Specifically, there are likely many Anglophone scholars in Central Europe, and Asia scholars in North America and Europe. As such, referring to these groups by their ethnicity can be misleading, as an Korean scholar residing in a Canadian university is likely to face very different authorship pressures than if they were in their home country. My suggestion is to rephrase to say something like “scholars residing in Anglophone, Central European, and Asian countries…”.

- Regression tables: The authors should mention the reference level for each categorical variable, either included directly in the table (as a named row with no coefficients) or in the table caption. For example, it is clear that “Male” is the reference level for the gender variable, but it is not clear what the reference level for geography is, or job title.

- Fig 1: The decimal points for percentages on the Y-axis can be removed to make the graph more clean

- Fig 1: An optional suggestion—making the styles consistent between figures 1 and 2 could really make them more aesthetically pleasing and make the final manuscript look more professional.

- Fig 2: text is too small and difficult to read, the font-size should be increased as much as possible

- Fig 2: the graph would be easier to read if there were borders around the bars that clearly delineated them from each other.

- Fig 2: This is only a suggestion, but this plot could be more readable and impactful if the authors find a way to combine A & B into a single panel, along with C & D into a second panel. For example, the bars for identified share of ghost authors could be shown, and a single point, or a line, to mark the perceived share. There are several ways this could be done, it is simply that comparing histograms side-by-side is more difficult than comparing within a single panel.

- Supplemental texts: It wold be preferred to upload the supporting information as a .pdf rather than a .docx. .pdfs are more universal, so by using it you are ensuring that the files are more accessible to whomever need them.

7. PLOS authors have the option to publish the peer review history of their article (what does this mean?). If published, this will include your full peer review and any attached files.

Reviewer #2: No

Reviewer #3: No

---

## [Author Response · Author response to Decision Letter 1]

23 Mar 2022

We provide detailed responses to the reviewers in two seperate documents.

---

## [Editor Report · Decision Letter 2]

7 Apr 2022

And the credit goes to … - Ghost and honorary authorship among social scientists

PONE-D-20-30365R2

Dear Dr. Pruschak,

We’re pleased to inform you that your manuscript has been judged scientifically suitable for publication and will be formally accepted for publication once it meets all outstanding technical requirements.

Kind regards,

Alberto Baccini, Ph.D.

Academic Editor

PLOS ONE
---

## [Editor Report · Acceptance letter]

20 Apr 2022

PONE-D-20-30365R2 

And the credit goes to … - Ghost and honorary authorship among social scientists 

Dear Dr. Pruschak:

I'm pleased to inform you that your manuscript has been deemed suitable for publication in PLOS ONE. Congratulations! Your manuscript is now with our production department. 

Kind regards, 

on behalf of

Prof. Alberto Baccini 

Academic Editor

PLOS ONE